# In Situ Monitoring of Aptamer–Protein Binding on a ZnO Surface Using Spectroscopic Ellipsometry

**DOI:** 10.3390/s23146353

**Published:** 2023-07-13

**Authors:** Adeem Alshammari, Harm van Zalinge, Ian Sandall

**Affiliations:** Department of Electrical Engineering & Electronics, University of Liverpool, Liverpool L69 3GJ, UKvzalinge@liverpool.ac.uk (H.v.Z.)

**Keywords:** aptamers, ellipsometry, biosensors

## Abstract

The dissolution of zinc oxide is investigated using spectroscopic ellipsometry to investigate its suitability as a platform for biosensing applications. The results indicate that once the ZnO surface has been functionalised, it is suitably protected, and no significant dissolving of the ZnO occurs. The binding kinetics of the SARS-CoV-2 spike protein on aptamer-functionalised zinc oxide surfaces are subsequently investigated. Values are extracted for the refractive index and associated optical constants for both the aptamer layer used and the protein itself. It is shown that upon an initial exposure to the protein, a rapid fluctuation in the surface density is observed. After around 20 min, this effect stabilises, and a fixed increase in the surface density is observed, which itself increases as the concentration of the protein is increased. This technique and setup are demonstrated to have a limit-of-detection down to 1 nanomole (nM) and display a linear response to concentrations up to 100 nM.

## 1. Introduction

Rapid biological sensing is becoming a key requirement and need for modern healthcare systems. Methods such as polymerase chain reaction (PCR), mass spectrometry, and NMR have all been extensively used for medical diagnostics over the past 50 years [1,2,3,4]. However, these approaches are all relatively slow (often requiring hours to days to generate a result), expensive to perform (due to both equipment and the need for highly trained staff), and often require sensitive sample preparation that may require dedicated instruments. Over the past decade, biosensors have emerged as an attractive alternative means of diagnosis, offering rapid detection, compactness, ease of use, and low cost, allowing for mass production and use at the point of care [5,6,7,8]. 

A wide range of biosensor methods and configurations have been demonstrated, utilising optical, electrical, and electrochemical modes of operation [9,10]. A common requirement for these different types of biosensors is to capture and immobilise the target of interest onto the sensor. Over the past decade, aptamers that comprise short strands of DNA, RNA, or peptides have proven an attractive alternative to more traditional antibodies, given their stability against pH and temperature changes, enabling target proteins or molecules to reside on the sensing surface [11,12]. This quality has led to the demonstration of a number of aptamer-based biosensors utilising electrochemical-, optical-, and electrical-based detection methods [13,14].

Regardless of the sensing method used in such aptamer-based sensors, the underlying change in the signal is driven by the change in concentration of the target protein or molecule on the surface of the active area of the sensor. As such, it is critical to have a good understanding of the dynamics of the target on the surface. Spectroscopic ellipsometry (SE) can be used as a real-time, accurate, and rapid measurement technique to optically study biomolecular interactions on solid surfaces in both in situ and ex situ environments [15,16]. The SE method is based on optical changes in the amplitude ratio Psi (ψ) and the phase difference Delta (∆) in terms of wavelength on thin film surfaces including adsorption and bio-immobilisation, where the sensing operation is calculated by the value of the surface mass density of the adsorbed target.

Due to its global importance, a large number of research groups have demonstrated biosensors for the rapid detection of COVID-19 over the past three years, including our own work on using aptamer-functionalised devices [17,18,19,20]. However, to date there has been no investigation of the in situ binding of the COVID-19 spike protein to aptamers on a potential sensor surface, and hence, much of the dynamics and subsequent device optimisation remain unclear. 

A key component of any biosensor system is the active material used for the substrate. Zinc oxide (ZnO) is an attractive material for potential biosensors given its relatively low cost, ease of deposition as a thin film, potential as a recyclable/biodegradable sensor, and transparency over the visible spectrum [21,22,23]. These qualities have resulted in a large number of publications demonstrating ZnO-based biosensors, for example, for cancer diagnosis, cardiac diseases, and virus detection [6,24]. However, it has been previously reported in the literature that ZnO is soluble in water, which will result in any potential sensor degrading over time [25].

In this work, we make use of ZnO as the substrate for the aptamers to bind to. We present a study looking into the solubility of ZnO during functionalisation. We will then show that the functionalised ZnO substrate can act as a sensor detecting spike proteins using spectral ellipsometry.

## 2. Materials and Methods

Nominally 80 nm ZnO films were deposited onto silicon substrates using an RF magnetron sputter system (NanoPVD, Moorfield, Knutsford, UK). The deposition process started with cleaning the Si substrates (SSP, 1–10 ohm-cm), cut into pieces with 2 × 4 cm^2^ dimensions. The Si substrates were cleaned with acetone, IPA, and ethanol for 5 min each then rinsed with DI water and dried with nitrogen gas. The substrates were placed in a sputter chamber with a ZnO target of 99.99% purity (Testbourne Ltd., Basingstoke, UK), and the chamber was evacuated until an initial pressure of 4.3 × 10^−7^ mBar. The conditions for the plasma during deposition were an Ar gas flow: 3.5 sccm, RF power of 52 watt, resulting in a growth rate of 0.025 nm/s at ambient temperature.

In this experiment, covalent immobilisation is required to modify the ZnO surface using an amino functional group. All chemicals were purchased from Merck and used as received, unless otherwise stated. All aqueous solutions were prepared in DI water (resistivity 18.2 mΩ cm) purified using a Direct-Q 3 water purification system (Merck/Millipore, Feltham, UK). The samples were immersed in a 3% APTES ((3-Aminopropyl) triehoxysilane) in absolute ethanol (ThermoFisher Scientific, Loughborough, UK) solution at 80 °C for 2 h then rinsed with DI water and dried under N_2_. Next, the ZnO samples were immersed in 2% glutaraldehyde in water at room temperature for 1 h then rinsed with DI water and dried. Then, 400 µL of 100 nM aptamer concentration in a PBS solution (137 mM sodium chloride, 2.7 mM potassium chloride, 10 mM phosphate buffer pH 7.4) were added to the sample surface and kept in a covered Petri dish at 37 °C for 2 h, followed by rinsing with PBS and DI water. There is a possibility that not all aldehyde groups provided by the glutaraldehyde bound to an aptamer, and, as such, they would provide sites that potentially can bind non-specifically to any amine group present in the proteins. To prevent this phenomenon, following the aptamer functionalisation, the samples were immersed in a PBS solution containing 80 mM glycine for 60 min at room temperature. They were subsequently rinsed in PBS and dried in an N_2_ atmosphere to remove excess material as well as any water in the layers. The functionalisation process is shown schematically in Figure 1. Afterwards, the spike protein (Recombinant SARS-CoV-2 spike protein (500 µg, 0.8 mg/mL), purchased from Cambridge Biosciences (Cambridge, UK)) was added in concentrations from 12.5 fM to 125 nM into the liquid cell during the SE measurements. The aptamer sequence used in this work was 5′-CAGCACCGACCTTGTGCTTTGGGAGTGCTGGTCCAAGGGCGTTAATGGACA-3′ with an amine group attached to the 5′ end, which has previously been reported to bind to the spike protein of the SARS-CoV-2 virus [26].

Optical measurements were performed using a spectroscopic ellipsometer (M-2000^®^ from J. A. Woollam Co., Inc., Lincoln, NE, USA) with a wavelength range of 245–1700 nm, and a 75° incidence angle was used for the measurements in this work. All measurements were acquired and analysed using CompleteEase^®^ analysis software, version 5.08, from J. A. Woollam Co., with which an optical model could be generated to model each layer using a Cauchy model [27]. The Cauchy transparent model dispersion functions best when the material has no or very low optical absorption in the visible spectral range. Most materials typically have a normal dispersion, which is characterised by a monotonously decreasing refractive index with increasing wavelength. The Cauchy formula for transparent dispersion for optical constants *n* and *k* can be expressed as:(1)$$n\left(\lambda \right)=A+\frac{B}{\lambda^{2}}+\frac{C}{\lambda^{4}},$$
(2)k(λ)=0,
where (*A*) is a dimensionless parameter, (*B*) affects the curvature and the amplitude of the refractive index for medium wavelengths in the visible spectral range, and (*C*) affects the curvature and amplitude for the UV range. The extinction coefficient is assumed to be equal to 0 for all wavelengths studied. Initial measurements were performed on deposited ZnO films to confirm the quality of the deposited layers.

Figure 2 shows the measured values of *ψ* and ∆ from the ellipsometer, both measured as a function of wavelength for bare ZnO film, the film after the aptamers were attached, and ZnO film upon exposure to spike proteins.

To extract the relevant information, such as refractive index and layer thicknesses, the spectra in Figure 3 undergo a fitting procedure using CompleteEASE. Firstly, scans and fits are performed on a reference piece of the underlying silicon substrate; for this purpose, library models are used for both the silicon and native oxide present on top. After this procedure, a scan is undertaken on a sample containing a sputtered ZnO layer; for the fitting of these spectra, the same parameters are used for the silicon and native oxide as determined from the reference, and a Cauchy model is used to fit the ZnO (as discussed in Section 2). This process is then repeated as each layer is formed on the sample, i.e., the parameters from the scan are used for those layers, and a Cauchy fit is performed for the new layer. The fitting procedure showed the thickness of the ZnO layer is (82.17 ± 0.04) nm, with a refractive index of 1.94 at 632 nm; the observed refractive index agrees well with the expected value from previous reports [27,28]. 

Following this verification of the system, subsequent in situ measurements were performed to establish both the dissolution rate of the ZnO film and the aptamer–protein binding dynamics. For this purpose, samples were placed in an aluminium liquid cell holder (5 mL capacity with a horizontal window, 75° angle of incidence), with a measurement scan taken continuously in real time. To investigate the change in the surface as the concentration of protein increases, various concentrations (12.5 fM to 125 nM) were injected into the liquid cell with real-time monitoring of absorption profile changes.

## 3. Results

It has previously been reported that ZnO can have a high dissolution rate in aqueous media [25], depending upon the pH, which may limit its potential as a platform for biosensing. The dissolution rate may critically affect the experiment at two stages. Firstly, during the fabrication and functionalisation procedure, it is possible that the ZnO film may be significantly dissolved and hence compromise its performance. For the functionalisation routine used in this work, the glycine step that utilises a low pH aqueous solvent is of particular concern. Secondly, the dissolution rate can affect the experiment during in situ measurements when the ZnO sensor is placed in a liquid cell filled with a PBS buffer solution, as continuous or long-time detection of the protein will be impacted by changes in the ZnO thin film thickness. However, there is also the possibility that once functionalisation has occurred, the ZnO film will in effect be encapsulated by the subsequent layers and as such be protected from any further dissolution. To investigate this effect, we have firstly undertaken in situ measurements on bare ZnO layers upon immersion in DI water and aqueous PBS. The software recorded spectra every second over a 60 min interval. Figure 3 shows the calculated thickness from these spectra every 10 min.

The dissolution rate in both cases is seen to be relatively high and as such would result in an unstable sensor due to substrate dissolution. The same measurement process was then performed on ZnO after each step of the functionalisation process, with the results also shown in Figure 2. The dissolution rate can be seen to be significantly reduced directly after the silanisation step, indicating that the silane layer forms an isolating layer on top of the ZnO surface. Although there is a slight increase in the dissolution for subsequent steps, it should be noted that the fitting of these layers becomes more complex at each functionalisation step as there is no longer just a ZnO layer followed by a single organic layer. As such, it may be that we are actually observing changes in the orientation/conformity of the differing layers during these steps rather than the ZnO dissolving. This possibility is further supported by noting that there seem to be at least two regions in Figure 2 for the glutaraldehyde, aptamer, and glycine steps; there is an initial decrease in thickness until around 30 min, followed by a levelling of the thickness, supporting the idea that during the first 30 min, the layers are optimising their orientation in order to minimise the potential energy before a steady state is achieved, while the underlying ZnO remains unchanged. Overall, dissolution is significantly less than 1 nm over a period of 1 hr, indicating that ZnO is a suitable medium for the target protein to bind to and will provide stable sensing for at least 1 h.

The spectra of Psi and Delta, shown previously in Figure 2, indicate a clear shift in the spectral shapes after the aptamers have been attached and again after protein binding has occurred. From the fitting of these spectra, we have calculated the refractive index of the differing layers after each step of the functionalisation process as well as for the final bound protein layer. Further Delta and Psi plots for these differing layers are given in the Appendix A. The resultant plots of the refractive index as a function of wavelength are shown in Figure 4. The refractive index for each layer has a similar initial large drop in refractive index, followed by a more gradual decline as the wavelength continues to increase. This analysis was also undertaken for lower concentrations of the spike protein, with similar refractive index curves calculated each time, although there was a higher uncertainty on these figures, due to the reduced surface coverage of the protein. Accurate knowledge of these parameters is important to enable accurate design and modeling of appropriate biosensors, and, to our knowledge, these are the first reported values for the spike protein and this aptamer sequence.

Spectroscopic ellipsometry has limitations in the modelling of biolayers to determine the thickness of the resultant layer(s) due to the formation of bioconjugates onto a modified surface where such a system has unknown optical properties. The model used to fit the experimental data assumes that the layer is uniform. The adsorption of the protein onto the surface as well as the underlying layers are far from uniform, creating issues with the fitting. In recent years, there has been significant progress in determining the optical parameters of thin films and polymer-based coatings [29,30]. However, currently the most common approach is to simplify measurements of the resulting thickness by effectively evaluating the density of the layer of the absorbed proteins. The obtained thicknesses and refractive indices from the protein adsorption process can then be transformed to a surface mass density using the De Feijter equation [31],
(3)$$\tau =d_{f}\left(n_{f}-n_{m}\right)\left(\frac{d_{c}}{d_{n}}\right)$$

In this equation, *d_f_* is the film thickness in nm, *n_f_* the refractive index, *n_m_* the aqueous medium refractive index, and *d_c_/d_n_* is an increment in the refractive index, assumed to be constant, approximately 0.183 mL g^−1^ in this instance [32].

We have undertaken this analysis on spectra recorded every minute with the samples placed in situ, using a wavelength of 625 nm, with the resultant surface mass density based upon the de Feijter equation, shown in Figure 5. For each measurement run, the sample is placed into the liquid cell then filled with PBS, and measurements are taken for 10 min; after this time, the protein (dispersed in PBS) is added to the liquid cell. For the lowest concentrations of protein, no changes are observed; however, for concentrations above 1.25 nM, an initial sharp increase in the surface density is observed upon the addition of the protein. Over the following 15 min, a fluctuating density is measured that then decays to a steady state value. During this middle fluctuating region, it seems likely that an equilibrium is being established, whereby proteins are binding and then being released until they are captured in the preferential configuration (lowest energy arrangement). During this period, the density of protein on the surface is in a high degree of flux, resulting in a changing signal. After approximately 15 min, a stable regime is entered, indicating that the proteins have now bound to the aptamers in the preferential configuration, forming a stable and constant surface density, ultimately resulting in a stable configuration on the surface. This result therefore also demonstrates a long-term binding of the protein to the aptamer. It is important to note this behaviour, as it suggests that any sensor using aptamers to bind the target will likely provide inaccurate and inconsistent results if measurements are taken during the initial dynamic phase but that after this point, the system would be stable. These results indicate that any sensor based on the attachment of proteins to aptamers would require a minimum settle time of 10 to 20 min to provide quantifiable results [15,33].

Figure 6 shows the stabilised surface density (defined as the density after a total time of 35 min) for different concentrations of the spike protein. For concentrations from 125 fM up to approximately 125 pM, no change in the density is observed, outside of the experimental variation, before an increase in the density is observed for higher protein concentrations. As can be observed in Figure 6, a linear fit between the concentrations of 0.1 nM to 100 nM (shown in red in Figure 6) yields a gradient of 0.78 ± 0.19, indicating a near linear increase in the surface density as the in-liquid concentration is increased. The errors given in this figure result from the propagation of the errors generated from the initial refractive index and thickness extraction from the Cauchy model; further details of these results are given in the Appendix A. From a physical point of view, a linear relationship seems reasonable as this indicates that a factor-of-10 increase in the concentration in the liquid results in a factor-of-10 increase in the surface density. A similar result is obtained if the analysis is repeated at different wavelengths. 

To investigate the selectivity of the aptamers in terms of specific binding to spike proteins, we have repeated the in situ analysis in which the spike protein has been replaced by bovine serum albumin (BSA) as a control. The result for adding 125 nM of BSA is shown in Figure 7, along with the original spike protein result at the same concentration (from Figure 5) as a reference. It can again be seen that upon the addition of the protein, there is a sudden rise, followed by a large random fluctuation in the calculated surface density, in a similar manner to what was originally seen for the spike protein. However, whereas for the spike protein this fluctuation settles after a period to give a constant density, indicating that stable long-term aptamer–protein binding has occurred, here, this settling does not occur. This result suggests that in the case of BSA, a constant state of rapid capture and quick release remains between the protein and the aptamers. This state results in a rapidly fluctuating surface density even after extended periods of time, which is in contrast to the result previously obtained for the spike protein. 

## 4. Discussion

These results have shown that spectral ellipsometry is a valid technique to verify that protein–aptamer binding has occurred. The results have demonstrated that for any such aptamer-based sensor, an initial wait time is required to enable the aptamer–protein to reach a stable steady state equilibrium in terms of protein binding and release. The results of this work have also enabled us to extract the refractive index for both the aptamer sequence used here and the spike protein itself; future optimisation of any aptamer-based sensor for COVID-19 detection would require detailed modelling and simulation, as such accurate knowledge of these parameters is critical. The results have enabled us to determine a relationship between the concentration of protein in the liquid sample and the resultant surface density of that protein on the surface. This relationship is important to enable sensors with a low limit of detection to be designed in the future.

Finally, this work has highlighted the severe limitations of spectral ellipsometry as a sensing technique itself when used in situ. While the work presented here has demonstrated that ellipsometry can be utilised to verify if aptamer binding has occurred, to provide insight into the timescales of binding, and to allow a relationship between the in-liquid protein concentration and the resultant surface density, it is not a suitable technique itself for the realisation of a biosensor. This unsuitability is due to the fact that to be able to verify that a target protein is present, a time-dependent measurement would be required to ensure that a steady state equilibrium had been reached in the signal indicating the formation of a protein layer. This requirement would significantly increase the cost and complexity of any such sensor system. Furthermore, in any real-world biological sample, there would be many proteins present apart from the target, and as indicated by the tests on BSA, this condition would result in a large noise floor, making definitive detection problematic if not impossible. Any such limitations are, however, due to the measurement technique and not a property of either the aptamers or ZnO, both of which have been shown elsewhere in the literature to enable the realisation of biosensors with high selectivity and sensitivity. For example, our previous work with this aptamer sequence has shown the operation of electronic-based biosensors capable of operating in a complex protein matrix with low sensitivity [17], while many other groups have demonstrated ZnO-based biosensors capable of selective detection.

## Figures and Tables

**Figure 1 sensors-23-06353-f001:**
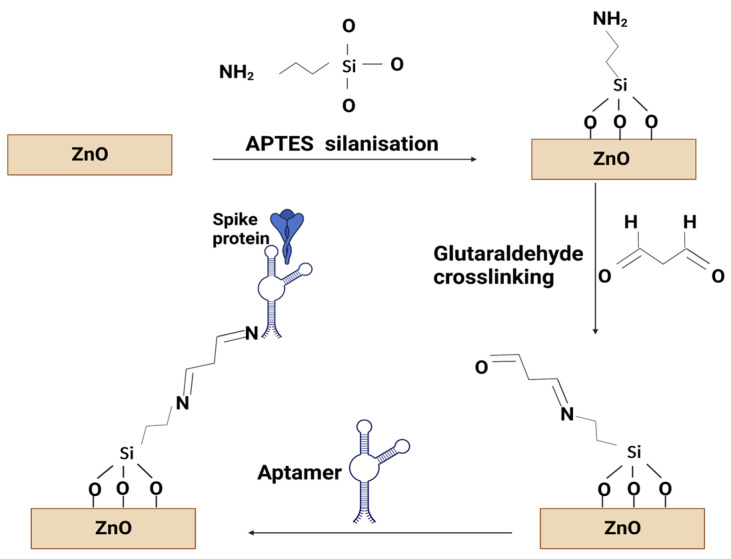
Schematic of the functionalisation of a ZnO surface.

**Figure 2 sensors-23-06353-f002:**
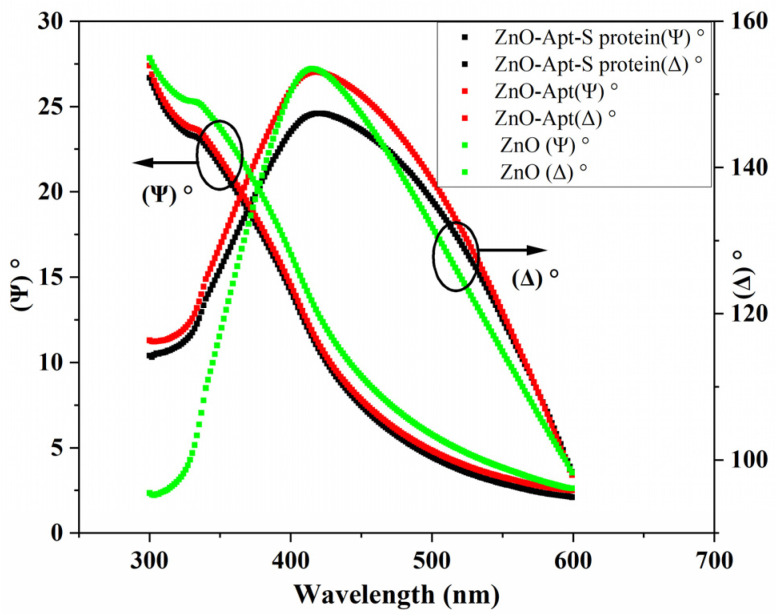
Ellipsometric parameters *ψ*(*λ*) and ∆(*λ*) of ZnO for a bare ZnO thin film (black curves), the aptamer-functionalised layer on a ZnO surface (red curves), and a spike protein layer immobilised on the ZnO surface (green lines).

**Figure 3 sensors-23-06353-f003:**
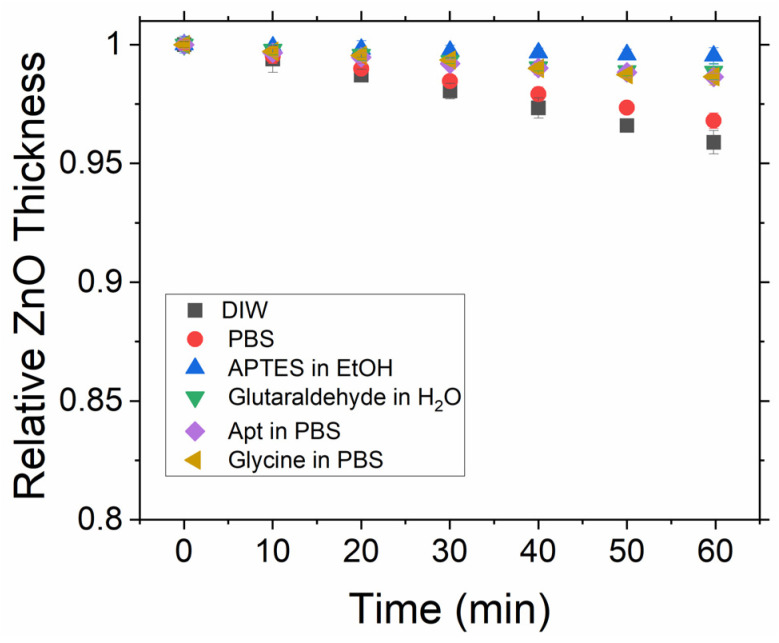
Time monitoring of ZnO thickness in the liquid cell filled with DIW and 0.1 M PBS before surface functionalisation. In addition, thickness monitoring after functionalisation steps with the associated liquid media.

**Figure 4 sensors-23-06353-f004:**
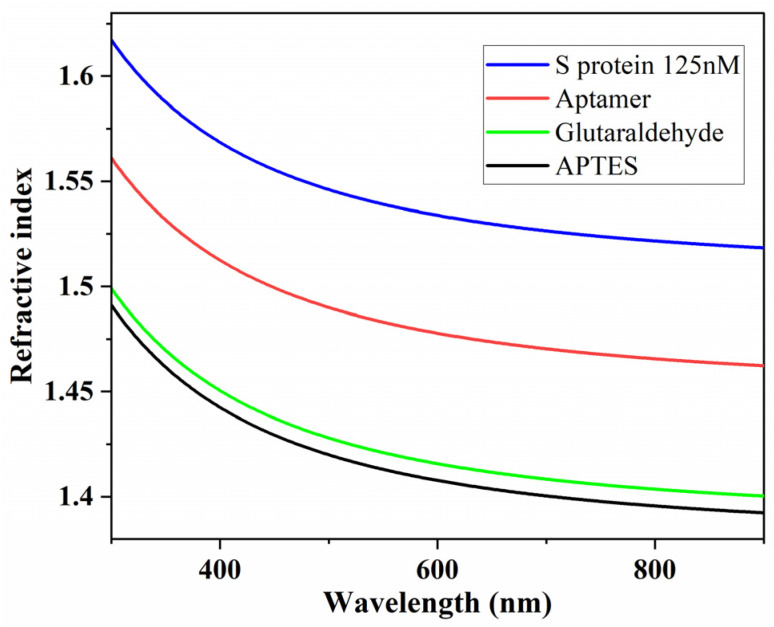
Extracted refractive indices, after all functionalisation steps, of the aptamer layer (red line) and of the spike protein layer (blue line).

**Figure 5 sensors-23-06353-f005:**
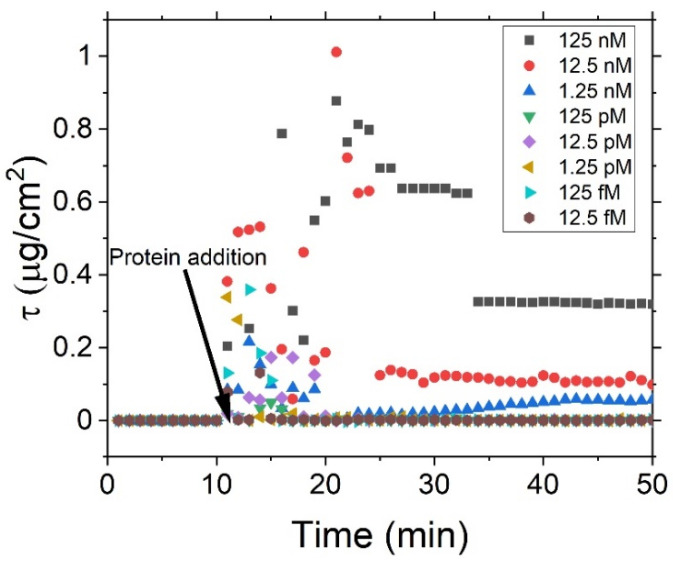
Time monitoring of the surface density for various in-liquid protein concentrations.

**Figure 6 sensors-23-06353-f006:**
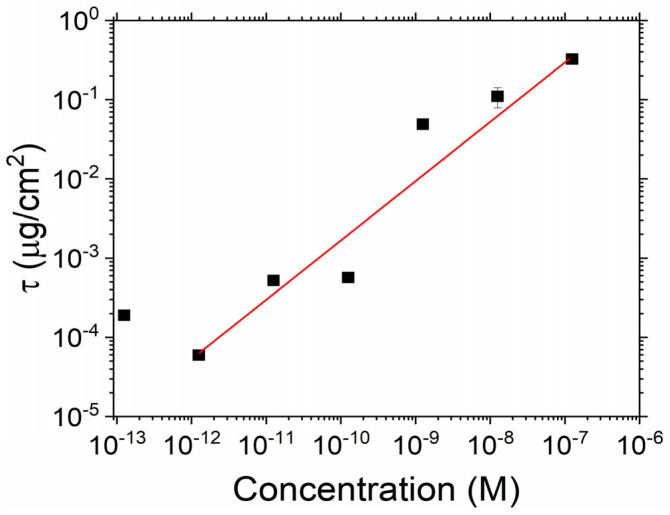
Surface mass density ratio of adsorbed protein as a function of the in-liquid protein concentration, taken 35 min after the protein was introduced.

**Figure 7 sensors-23-06353-f007:**
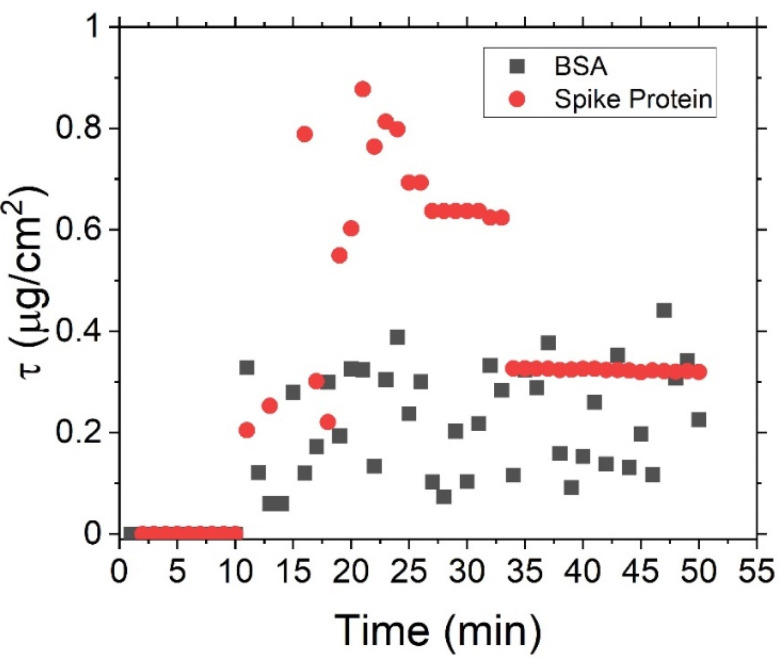
Time monitoring of 125 nM spike protein concentration along with a BSA reference sample.

## Data Availability

The data used within this study are available from the authors.

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
