# Peer review of "In Situ Monitoring of Aptamer–Protein Binding on a ZnO Surface Using Spectroscopic Ellipsometry"

_sensors, 2023, doi:10.3390/s23146353_

Round 1

Author Response

We thank the reviewers for their time and careful review of our manuscript. On the whole we feel they have made valuable contributions which have helped to improved the overall quality of the paper. Please find below a summary of the points raised by each reviewer and our subsequent response to these (in red). We have also provided a marked copy of the final manuscript with all changes highlighted.

Reviewer 1

In this manuscript, Adeem Alshammari et al. firstly demonstrated that the dissolution rate of ZnO was significantly reduced after functionalization by monitoring the thickness of ZnO in different liquid mediums. And then they made the conclusion that the ZnO is a suitable platform for biosensing and it can provide stable sensing for at least 1 hour. Next, the authors investigated the binding kinetics of the spike protein to the aptamers using spectroscopic ellipsometry. And the measurement results showed that after a period (about 15 minutes), a steady value of surface density was achieved when the concentration is above 1.25 nM. At last, the authors studied the selectivity of the aptamers by using BSA as a control.

The writing style of this paper is engaging and concise, making it easy to follow the authors’ line of reasoning. The discussion section of this paper is relatively strong. The authors have analyzed their results, addressed potential limitations, and provided insightful interpretations. However, this work is lack of novelty, and was not thoughtfully studied. I would suggest make a major revision for publication. Specific comments are given below:

  1. For a sensor, accuracy and reliability are key parameters. In this article, the authors showed that the ZnO can act as a sensor to detect the spike protein using ellipsometry, but only response time, and detection limit were studied, no other characteristics were provided.

The primary motivation of the manuscript was to utilise ellipsometry to

  1. Verify the stability of ZnO in terms of its disillusion rate both pre and post functionalisation to evaluate the potential of the substrate as a potential platform for biosensors
  2. Investigate the aptamer-protein binding dynamics and kinetics

As such wider investigation of ZnO and ellipsometery as a sensor was a secondary priority, however as noted by the reviewer we have still evaluated the detection limit and response time from this approach. The results shown in figure 7 also indicate the selectivity of this as a approach as a sensor. As such we have provided information on the key three parameters of any sensor system, namely: Selectivity, Sensitivity (both in terms of its limit and dynamic range) and Response Time.

  1. For figure 6, the interpretation in line 226 and 227 are not supported by the figure. And the statements made here should be more supported by other evidence or experiment results. The data in figure 6 shows the extracted surface density as a function of the in liquid concentration over the concentration range studied. For the initial concertation range (~125 fM to 125 pM) there is little change in the measured density. In fact there is an initial decrease in the measured density followed by a slight increase and then a more pronounced increase as the concentration goes above 125 pM. There is no physical reason for the surface density to decrease as the in liquid concentration is increased and as such we believe this is just an artefact of the technique and represents larger experimental uncertainty in the technique. As such we feel that only the increase in density after 125 pM is real. The discussion on lines 226 and 227 have been clarified to emphasise this point.
  2. As the authors stated, “in any real-world biological sample there would be many proteins apart from the target present and as indicated by the tests on BSA this would result in a large noise floor making definitive detection problematic.” So without overcoming this interference issue from other substances, how does the functionalized ZnO become a valuable platform for biosensors?

This statement and conclusion does not refer to the potential of Zinc Oxide as a substrate platform but rather is a comment on the limitation of Specto-Ellipsometery as a diagnostic tool. As has been reported by a variety of groups ZnO can be used as a biosensor utilising a range of detection techniques and this work does not contradict the potential of ZnO as substrate for biosensors. What these results have demonstrated is that while Spectro-Ellipsometery is a powerful tool to study aptamer-protien binding, verify selectivity of the aptamers and enable a relationship between in liquid protein concentration and resultant surface density to be determined, ellipsometery is not in itself a viable technique for a biosensor. We have added some further discussion in the final paragraph to discuss and clarify this.

  1. Could the authors explain why a wavelength of 625 nm was selected to conduct the analysis?

As noted in the paper, similar results are seen regardless of the wavelength we selected. We decided to utilise 625 nm, due to it being close to the middle of the spectrum that was measured and not in a region where the refractive index was rapidly changing as a function of wavelength.

Reviewer 2 Report

The paper examines the potential of spectroscopic ellipsometry as a platform for bio-sensing applications. It delves into the investigation of binding kinetics between the SARS-CoV-2 spike protein and aptamer-functionalized zinc oxide surfaces. These studies hold immense relevance, and major revisions are necessary before the work can be considered for publication in the journal "Sensors."

1.       The choice of information presented in Figure 4 is surprising. The caption to the figure states that the black line refers to the refractive index of the buffer solution (PBS), while in the tab to the figure, this line is marked with the caption "after silanization"???. That is, before glutaraldehyde??? Judging by the value of the specified refractive index, this curve really refers to the PBS buffer solution. However, what is the purpose of presenting this parameter in the figure? It would be much more interesting to learn the optical parameters of the APTES and glutaraldehyde layers, since changes in the spectrum of ellipsometric parameters D,Y occur at every stage of the surface modification, in particular also after the APTES silanization and glutargenization. This information would be useful in interpreting the results of ZnO film solubility studies at different stages of surface modification.

2.       In line 162 we read "for the glutaraldehyde, aptamer, and glycine steps." It is not clear what glycine step we are talking about.

3.       On page 6, lines 190 and 194, instead of "absorption of the protein" and "protein absorption", it should obviously be "adsorption of the protein" and "protein adsorption".

4.       It is clear for what reasons the authors use instead of the thickness and refractive index of the protein layer and its surface density, calculated using the de Feijter equation. However, the work would look more advantageous if at least the limits, in which the thickness of the protein layer, as well as its refractive index at 625 nm, are changed, were given.

By the way, in line 202, the author of the formula changed his last name (de Freijter ??).

5.       The authors also mention the limitations of the ellipsometric technique "in the modeling of biolayers to determine the thickness of the resulting layer(s) due to the formation of bioconjugates onto a modified surface where such a system has unknown optical properties." However, it should be noted that the technique of ellipsometric measurements continues to actively develop and various authors are making attempts to improve it. In particular, in this aspect, it is worth mentioning the works that show the new possibilities of the ellipsometric technique in the study of ultra-thin polymer and copolymer coatings to determine their composition based on ellipsometry data. Given the perspective of further research on biosensors based on the ellipsometric method, it is advisable to add the following references to the list:

https://doi.org/10.1039/C7SM02285A

https://doi.org/10.1364/OE.25.027077

Minor editing of English language required.

Author Response

We thank the reviewers for their time and careful review of our manuscript. On the whole we feel they have made valuable contributions which have helped to improved the overall quality of the paper. Please find below a summary of the points raised by each reviewer and our subsequent response to these (in red). We have also provided a marked copy of the final manuscript with all changes highlighted.

Reviewer 2

The paper examines the potential of spectroscopic ellipsometry as a platform for bio-sensing applications. It delves into the investigation of binding kinetics between the SARS-CoV-2 spike protein and aptamer-functionalized zinc oxide surfaces. These studies hold immense relevance, and major revisions are necessary before the work can be considered for publication in the journal "Sensors."

  1. The choice of information presented in Figure 4 is surprising. The caption to the figure states that the black line refers to the refractive index of the buffer solution (PBS), while in the tab to the figure, this line is marked with the caption "after silanization"???. That is, before glutaraldehyde??? Judging by the value of the specified refractive index, this curve really refers to the PBS buffer solution. However, what is the purpose of presenting this parameter in the figure? It would be much more interesting to learn the optical parameters of the APTES and glutaraldehyde layers, since changes in the spectrum of ellipsometric parameters D,Y occur at every stage of the surface modification, in particular also after the APTES silanization and glutargenization. This information would be useful in interpreting the results of ZnO film solubility studies at different stages of surface modification.

Thank you for this suggestion, we have amended figure 4 to now show the refractive index profile of the APTES layer, Glutaraldehyde, Aptamer and protein and made it clearer as to what each graph s showing. The discussion areound this figure has been updated to make it clearer what is being shown and to discuss the results.

  1. In line 162 we read "for the glutaraldehyde, aptamer, and glycine steps." It is not clear what glycine step we are talking about.

During the functionalisation / fabrication there is a possibility that not all aldehyde groups provided by the glutaraldehyde bound to an aptamer and as such, they would provide sites that potentially can bind non-specifically to any amine group present in the proteins. To prevent this, following the aptamer functionalisation, the samples were immersed in a PBS solution containing 80 mM glycine for 60 min at room temperature. They were subsequently rinsed in PBS and dried in an N2 atmosphere to remove excess material as well as any water in the layers. A statement has been added to the materials and methods section to clarify this.

  1. On page 6, lines 190 and 194, instead of "absorption of the protein" and "protein absorption", it should obviously be "adsorption of the protein" and "protein adsorption".

Many thanks for spotting these, they have now been corrected.

  1. It is clear for what reasons the authors use instead of the thickness and refractive index of the protein layer and its surface density, calculated using the de Feijter equation. However, the work would look more advantageous if at least the limits,in which the thickness of the protein layer, as well as its refractive index at 625 nm, are changed, were given.

We have included a short discussion on the errors for the extracted Refractive Index and Thickness values both in the main text and also provided a table of errors in supplementary information (S2)

By the way, in line 202, the author of the formula changed his last name (de Freijter ??). – Thanks corrected

  1. The authors also mention the limitations of the ellipsometric technique "in the modeling of biolayers to determine the thickness of the resulting layer(s) due to the formation of bioconjugates onto a modified surface where such a system has unknown optical properties." However, it should be noted that the technique of ellipsometric measurements continues to actively develop and various authors are making attempts to improve it. In particular, in this aspect, it is worth mentioning the works that show the new possibilities of the ellipsometric technique in the study of ultra-thin polymer and copolymer coatings to determine their composition based on ellipsometry data. Given the perspective of further research on biosensors based on the ellipsometric method, it is advisable to add the following references to the list:

https://doi.org/10.1039/C7SM02285A

https://doi.org/10.1364/OE.25.027077

Many thanks, these references have been added when introducing the de Feijter method.

Reviewer 3 Report

The manuscript of Alshammari et al. reports on the fabrication and testing of the sensing device comprised of ZnO surface functionalized by aptamer and able to specifically detect SARS-CoV-2 spike protein by means of ellipsometric studies. Despite the performed research being of high interest, the presented manuscript suffers from the lack of sufficient explanations on the synthesis and characterization procedures, which make it premature to be published. Some details are below.

1.      The name of a particular aptamer used in the study should be introduced to the text. Two different aptamers were synthesized and studied in Ref. 26, so what the aptamer used in the present work is totally unclear

2.      Employing the antibody model in Figure 1 (bottom line) to illustrate aptamer will confuse the reader. An aptamer schematic model should replace it

3.      DNA aptamers do not have any -NH2 group at both 5’ and 3’ ends. What is the origin of the –NH2 group in the aptamer applied for the fabrication of the “ZnO-Aptamer” composite layer?

4.      The immobilization efficiency (for instance, in terms of the relation between the covered by aptamers and the total surface of ZnO particles) should be estimated. The covalent bonding between the glutaraldehyde and aptamer should be verified by any technique (FT IR, for instance). Otherwise, the use of the proposed modification method is not justified, aptamers can just be physisorbed on the surface of ZnO particles.

5.      The difference spectra for the ellipsometric parameters of ZnO-Aptamer substrated prior to and after being exposed to spike proteins should be added to Figure 3. Currently, no considerable difference in these parameters for the ZnO-Aptamers substrate and ZnO-Aptamers substrate with the spike protein is seen.

Authors should go carefully throughout the text and check punctuation. 

Author Response

We thank the reviewers for their time and careful review of our manuscript. On the whole we feel they have made valuable contributions which have helped to improved the overall quality of the paper. Please find below a summary of the points raised by each reviewer and our subsequent response to these (in red). We have also provided a marked copy of the final manuscript with all changes highlighted.

Reviewer 3

The manuscript of Alshammari et al. reports on the fabrication and testing of the sensing device comprised of ZnO surface functionalized by aptamer and able to specifically detect SARS-CoV-2 spike protein by means of ellipsometric studies. Despite the performed research being of high interest, the presented manuscript suffers from the lack of sufficient explanations on the synthesis and characterization procedures, which make it premature to be published. Some details are below.

  1. The name of a particular aptamer used in the study should be introduced to the text. Two different aptamers were synthesized and studied in Ref. 26, so what the aptamer used in the present work is totally unclear

The aptamer sequence has been added to the materials and methods

  1. Employing the antibody model in Figure 1 (bottom line) to illustrate aptamer will confuse the reader. An aptamer schematic model should replace it

We have updated an amended figure 1 accordingly

  1. DNA aptamers do not have any -NH2group at both 5’ and 3’ ends. What is the origin of the –NH2 group in the aptamer applied for the fabrication of the “ZnO-Aptamer” composite layer? - Naturally, unmodified DNA aptamers, the 5' and 3' ends of the DNA strands do not typically have NH2 groups, as stated. However, during the modification of DNA aptamers, Amino modification involves adding an amino group APTES + glutaraldehyde crosslinker. In this step, the composite layer of (APTES + glutaraldehyde crosslinker ) binds to 5' or 3' ends, and the other end binds to the hydroxyl group (-OH) of the ZnO surface. This is a common approach used in aptamer based sensors and has been relatively widely reported elsewhere.

  1. The immobilization efficiency (for instance, in terms of the relation between the covered by aptamers and the total surface of ZnO particles) should be estimated. The covalent bonding between the glutaraldehyde and aptamer should be verified by any technique (FT IR, for instance). Otherwise, the use of the proposed modification method is not justified, aptamers can just be physisorbed on the surface of ZnO particles.

In this work we are using sputtered 2D thin films of ZnO not a layer built from ZnO nano particles. The techniques and methods used in this work are standard techniques for the functionalisation of thin films with aptamers, which are accepted in the literature to give high surface coverage. The aptamer sequence used in this work has previously been demonstrated to bind to spike protein and the results here support that and provide further evidence in terms of selective binding. We do not have capability within our lab to verify the covalent bonding between glutaraldehyde and the aptamer, however again this is a commonly accepted procedure and process used by multiple groups. While there are a range of methods / approaches that could be used to attach the aptamers to the ZnO surface including just using physisorbed aptamers, the focus of this paper is to evaluate/study the aptamer-protein binding dynamics and not compare/contrast differing aptamer attachment procedures. While such a study would be of interest it is not within the aim or remit of this work.

  1. The difference spectra for the ellipsometric parameters of ZnO-Aptamer substrated prior to and after being exposed to spike proteins should be added to Figure 3. Currently, no considerable difference in these parameters for the ZnO-Aptamers substrate and ZnO-Aptamers substrate with the spike protein is seen.

The plots in figure 3 show the bare ZnO surface, the surface after aptamers have been attached and a further plot after the sample was exposed to in liquid protein. The labelling of Figure 3 has been updated to make this clearer and some further psi and delta plots have been included as supplementary information, to show the spectral shape at other intermediate steps. To aid the flow and discussion of the paper figure 3 has been moved to the methods section (now labelled as figure 2) and an enhanced discussion on the spectra and subsequent parameter extraction has been provided. Some further plots of delta and psi have also been included as supplementary information.

Round 2

Reviewer 1 Report

The authors have thoroughly revised the manuscript and answered the referee’s questions.

Reviewer 2 Report

The authors have answered all my comments since the paper can be accepted in its present form.

 Minor editing of English language required.

Reviewer 3 Report

The authors have adequately addressed all the comments. The manuscript can be published in a present form.